# The Benthic Quality Index to Assess Water Quality of Lakes May Be Affected by Confounding Environmental Features

**Angela Boggero** [1,*], **Silvia Zaupa** [1], **Roberta Bettinetti** [2], **Marzia Ciampittiello** [1] **and Diego Fontaneto** [1]

[1]  National Research Council-Water Research Institute (CNR-IRSA), Corso Tonolli 50, 28922 Verbania, Italy; silvia.zaupa@irsa.cnr.it (S.Z.); marzia.ciampittiello@irsa.cnr.it (M.C.); diego.fontaneto@irsa.cnr.it (D.F.)

[2]  Department of Human Sciences and Innovation for the Territory (DiSUIT), University of Insubria, Via Valleggio 11, 22100 Como, Italy; roberta.bettinetti@uninsubria.it

[*]  Correspondence: angela.boggero@irsa.cnr.it

**Abstract:** To assess if environmental differences other than water quality may affect the outcome of the Benthic Quality Index, a comparison of the application of four different methods (Benthic Quality Index—BQIES, Lake Habitat Modification Score—LHMS, Lake Habitat Quality Assessment—LHQA and Organisation for Economic Co-operation and Development—OECD) used to classify the lake ecological and hydro-morphological status of 10 Italian lakes was performed. Five lakes were natural and five were reservoirs belonging to both Alpine and Mediterranean Ecoregions. The 10 lakes were sampled using the Water Framework Directive compliant standardized national protocol, which includes sampling soft sediment in the littoral, sublittoral and deep layers along transects with a grab of 225 cm$^2$ during spring and autumn. The application of Generalised Linear Mixed Effect Models both at the lake level and at the single station of each lake highlighted that, at the lake level, no significant correlations existed between any couple of hydro-morphological, ecological and trophic status assessments, with each metric representing a different facet of human impact on the environment. At the single site level, we found significant effects of depth on the metrics of biodiversity. The best approximation of single-site macroinvertebrates diversity among the metrics of overall lake quality was with the LHMS, but not with the BQIES. Our hypotheses that lake macroinvertebrates assemblages depend also on other potential confounding variables of habitat degradation and intrinsic differences between lakes were confirmed, with depth playing a major role. Therefore, the assessment of lakes with different depths may produce different whole-lake BQIES values, only because of the effect of depth gradient and not because of differences in lake quality.

**Keywords:** grain-size; sediment; chemical analysis; macroinvertebrates; ecological status; Water Framework Directive; multimetric indices

## 1. Introduction

Before the launch of the Water Framework Directive (WFD, Directive 2000/60/EC), due to the extensive use of waters for indoor and outdoor purposes (e.g., hydropower generation, domestic, agricultural, industrial and recreation scopes), several of the aquatic ecosystems in Europe were heavily degraded, and many of them completely lost, sometimes even in an irreversible way [1,2]. Thus, the WFD is an important component in supporting the water sector in Europe, emphasizing the role of aquatic ecology in management decisions to protect an exhaustible resource [3]. Water resources management is based on a comprehensive understanding of ecosystem functions and interactions,

so a multi-parametric approach was needed to sustain water policy at the European level, considering future conservation and restoration actions [4].

In order to improve the quality of surface water bodies (lakes and rivers), specific studies focused on the implementation of monitoring and assessment methods across Europe. The most prominent are:

- Screening methods for Water Data Information in support of the implementation of the Water Framework Directive (SWIFT-WFD: www.swift-wfd.com) [5];
- Development and Testing of an Integrated Assessment System for the Ecological Quality of Streams and Rivers throughout Europe using Benthic Macroinvertebrates (AQUEM: www.aqem.de) [6];
- Standardisation of River Classifications: Framework method for calibrating different biological survey results against ecological quality classifications to be developed for the Water Framework Directive (STAR: www.eu-star.at/frameset.htm) [7];
- Relationships between the ecological and chemical status of surface waters (REBECCA: www.ymparisto.fi/eng/research/euproj/rebecca/homepage.html) [8];
- Water Bodies in Europe: Integrative systems to assess ecological status and recovery (WISER: www.wiser.eu) [9];
- Local hydro-morphology, habitat and RBMPs: new measures to improve ecological quality in South European rivers and lakes (INHABIT: www.life-inhabit.it);
- Managing aquatic ecosystems and water resources under multiple stresses (MARS: www.mars-project.eu) [10,11].

The evaluation of the ecological status of a lake requires an integrated approach that takes into account the effects on biota of different pressures encountered in lakes (eutrophication, acidification, general degradation, morphological alteration, etc.). As a general rule, the composition and the characteristics of the biotic communities of European waters, and the abiotic conditions influencing them, have become a primary focus to be analyzed in order to assess the quality of lakes and rivers as a whole. Since 2000, several biological assessment metrics, biotic indices or predictive models covering taxonomic, functional and trait-based approaches considering different pressures were developed by various countries in Europe [12–21], with the aim of improving management and conservation actions throughout Europe, and of harmonizing the classification of ecological status. Italy, after recognizing of the importance of eutrophication as a pressure impact on the national territory (nearly 41% of the Italian lakes were eutrophic [22]), decided to assess eutrophication impacts in lakes. This goal was reached using specific indices, ecological quality ratios and chemical and hydro-morphological status assessment to define reference conditions. Different metrics for phytoplankton, diatoms, macrophytes, macroinvertebrates and fish were developed within the remits of the WFD (for details on the Italian metrics adopted for each Biological Quality Element (BQE), see www.ise.cnr.it/wfd-en). Among such metrics, the Benthic Quality Index (BQIES) [23,24] considers the composition of the macroinvertebrates assemblages in order to assess the eutrophication levels of lakes. The index is based on a species-level approach for all benthic macroinvertebrates, mainly for chironomids and oligochaetes, co-dominating lake benthic communities. Then, in a second step, using quantile regression analysis, a rapid bio-assessment methodology of quality conditions has been set up to be submitted to the authorities responsible for water monitoring and to water managers [25]. The application of the rapid bio-assessment methodology has the objective of optimizing the sampling procedures of the national standardized protocol for monitoring lakes [26]. The aim is to support the environmental agencies responsible for the assessment of ecological quality in identifying entire lakes or parts of them that are altered, turning their attention to them and starting remediation actions. In 2018, the BQIES was finally accepted at the European level (UE Decision 2018/229) and became fully operational at the end of the same year.

The BQIES, like other indices, should reflect the effect of pollution on water quality, but also the effects of the physical, chemical, biological and biogeographic characteristics of each water body [27]. The explicit assessment of most of these environmental features is not currently compulsory in

the application of the BQIES [24], but may be used as a further support in the definition of high ecological status. We here want to demonstrate that the nature of the sediment, the lake depth and the water chemistry, and not only the trophic status, are factors affecting the outcome of the BQIES [28–31]. This paper tests the hypothesis that macroinvertebrates, sampled using a standardized methodology over a short period of time (spring to autumn of the same year) in both natural lakes and reservoirs in Italy, respond to a gradient of trophic state in which agriculture and animal husbandry are the predominant stressors, according to the current use of the BQIES, but also to environmental variables that could represent confounding factors for the strict application of the BQIES. To assess if environmental differences other than water quality may affect the outcome of the BQIES, we performed a comparison of the application of four different methods (BQIES, Lake Habitat Modification Score—LHMS, Lake Habitat Quality Assessment—LHQA, Organisation for Economic Co-operation and Development—OECD) used to classify lake ecological and hydro-morphological status. Then, we analyzed how the variability in species compositions of lake macroinvertebrates is related to geographic and local scale environmental factors, including sediment texture, sediment organic and inorganic matter content, hydro-chemical conditions and depth, in comparison with other indices of environmental degradation. The outcome of our analyses could be used to improve the application of the BQIES and to make it more compliant to the aims of the WFD, clarifying the importance of local environmental parameters not only as a support tool, but as a key means of characterizing the sediment on which life within them depends.

## 2. Materials and Methods

### 2.1. Study Area

The WFD states that all lakes with a surface area >0.5 km$^2$ should be monitored [22], but administrative regions, or parks or water managers, can decide to include smaller lakes deserving particular protection and safeguarding because of their strategic importance as a drinking water supply, their particular environmental value, or the peculiar character of the fauna and flora inhabiting them [32].

The lakes included in the current study belong to different types according to the EC Water Framework Directive classification system [33], and to two separate Ecoregions, together covering the meteorological conditions typical of the whole country. The lakes are classified into Alpine (AL) and Mediterranean lakes (ME) on the basis of the Ecoregion agreement [33]. The choice of this set of lakes and reservoirs was based on previous information about their trophic state, allowing us to cover a gradient of trophic state (from ultra-oligotrophy to hypertrophy) within which to test our hypotheses.

Six lakes are located in continental Italy (north-western side, Piedmont) and the remaining four in insular Italy (Sardinia) (Figure 1). They thus are subject to different meteo-climatic conditions. Five lakes are natural and five are reservoirs, of which four are placed in rural areas and one in natural settings (Table 1).

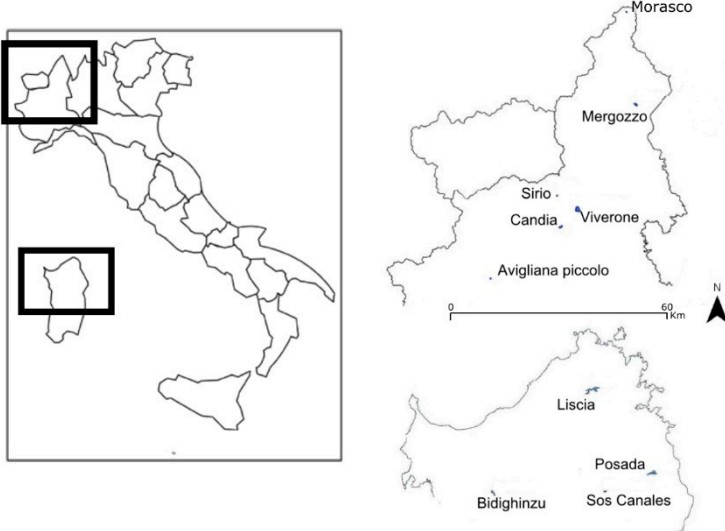

**Figure 1.** Distribution of considered administrative regions and sampled lakes in Italy.

**Table 1.** Administrative regions, lake names and types, ecoregions (AL = Alpine, ME = Mediterranean) and class types of lakes, and their respective main geographic and morphological information. Latitude and longitude, in WGS84 system, are expressed as DMS (Degrees, Minutes, Seconds); Altitude: m a.s.l.; Lake area: km$^2$; Mean and Maximum depth: m.

| Region | Lake Name | Lake Type | Class Type | Latitude N | Longitude E | Altitude | Lake Area | Mean Depth | Maximum Depth |
|---|---|---|---|---|---|---|---|---|---|
| Piedmont | Avigliana piccolo | Natural | AL-5 | 45°03′13″ | 07°23′30″ | 356 | 0.58 | 7.70 | 12.00 |
| Piedmont | Candia | Natural | AL-5 | 45°19′25″ | 07°54′43″ | 227 | 1.69 | 5.90 | 8.00 |
| Piedmont | Mergozzo | Natural | AL-6 | 45°57′23″ | 08°27′47″ | 194 | 1.82 | 45.60 | 73.00 |
| Piedmont | Morasco | Reservoir | AL-9 | 46°25′33″ | 08°23′48″ | 1814 | 0.57 | 31.00 | 50.00 |
| Piedmont | Sirio | Natural | AL-6 | 45°29′06″ | 07°53′05″ | 271 | 0.32 | 18.10 | 44.00 |
| Piedmont | Viverone | Natural | AL-6 | 45°24′05″ | 08°03′05″ | 230 | 5.78 | 22.50 | 50.00 |
| Sardinia | Bidighinzu | Reservoir | ME-2 | 40°33′24″ | 08°39′44″ | 330 | 1.50 | 8.40 | 30.00 |
| Sardinia | Liscia | Reservoir | ME-4 | 40°59′39″ | 09°14′37″ | 177 | 5.57 | 18.80 | 63.50 |
| Sardinia | Posada | Reservoir | ME-3 | 40°38′19″ | 09°36′28″ | 43 | 3.00 | 9.30 | 29.50 |
| Sardinia | Sos Canales | Reservoir | ME-5 | 40°33′17″ | 09°18′55″ | 709 | 0.30 | 19.70 | 47.50 |

*2.2. Lake Classification*

Water bodies (natural lakes and reservoirs) were sampled for biota and physical and chemical analyses in spring and autumn along transects connecting the littoral, sub-littoral and deep areas of each lake, according to the national protocol for macroinvertebrates sampling [26].

Four different methods were used to classify lake ecological and hydro-morphological status: OECD, LHMS, LHQA and BQIES. The methodology proposed by the Organisation for Economic Co-operation and Development (OECD [34]) was used to classify lakes on the basis of their trophic conditions. This metric is based on the hypolimnic oxygen concentrations in deep layers, on the mean values of total phosphorus (TP) at mixing, on mean values of chlorophyll *a*, and on mean annual values of transparency. TP, oxygen and chlorophyll *a* analyses were performed following Tartari and Mosello [35], and transparency was measured with a Secchi disc.

The summer application of the Lake Habitat Survey (LHS) was used to characterize the hydro-morphological conditions of each lake [36]. The LHS, based on a combination of habitat plot (Hab-Plots) observations, generates two main summary metrics: LHMS (Lake Habitat Modification Score), related to the degree of site modification, and LHQA (Lake Habitat Quality Assessment), to measure the diversity and naturalness of the lakes. Both metrics in the LHS assessment method, which surveys the terrestrial/aquatic ecotone, include quantitative descriptions of vegetation canopy, macrophytes composition and distribution, main littoral substrate and the presence of human impacts

on the shores and riparian zone [36]. High values of LHMS indicate high human modification and thus low ecological status, whereas high values of LHQA indicate high naturalness and thus high ecological status or high habitat quality.

The Benthic Quality Index (BQIES) was applied to the lakes as well. It is based on indicator weights attributed to each species, assuming that a species that is known to live preferably in high diversity sites is an indicator of a healthy environment, whereas a species that is known to be abundant in low diversity sites is an indicator of altered environments. Thus, high values of the index indicate high biodiversity and high ecological status [24].

*2.3. Sampling Methodology*

Biota and soft sediment were sampled with a grab (area = 225 cm$^2$); biota was sieved in the field through a net (mesh 250 µm), and stored with an aqueous solution (5%) of buffered formaldehyde. Water samples were collected by a Niskin bottle equipped with an overturning thermometer to obtain water parameters from the different sampling depths. All samples were brought to the laboratory for subsequent analyses.

Water features were measured for each sampling point in each lake according to Fornaroli and co-authors [25], and include both physical and chemical metrics: temperature, oxygen concentration, pH, conductivity, alkalinity, total phosphorus (TP) and total nitrogen (TN) (Table S1).

Sediment features were analyzed regarding chemistry and texture. For sediment chemistry, the water content, organic and inorganic matter and percentage of carbonates were measured via Loss On Ignition analysis (LOI 500 °C—[37]). For sediment texture, grain size was analyzed via Wentworth scaling (U.S. Standard) [38], allowing the separation of the sediment into many different fractions below 2 mm [39,40], and their classification according to their constituent parts (clay, silt and fine sand, expressed as percentages).

In the lab, sediment samples were sorted under a stereomicroscope to identify macroinvertebrates. The animals were separated into main groups, identified to species level when the presence of juveniles did not preclude it, and counted. The identification manuals used are those in use at the international and national levels ([41] for Chironomids; [42] for Oligochaetes, and [43] for the remaining taxonomic groups). Richness (number of taxa, considering mostly species, but also genera and families for minor taxonomic groups) and diversity (Shannon diversity index—SDI), representing the community structure and complexity, were calculated for each single-site sampling point.

The biological data for all taxa identified among the macroinvertebrates were used to apply the BQIES to each sampling point within a lake (BQIES$_{single-site}$—[24]). For each single-site sampling point we obtained biological and abiotic measurements.

In addition to the analyses performed at each sampling point, we also followed the current regulations, averaging values of all sampling points through space and time to obtain a mean annual BQIES$_{whole-lake}$ value for each lake.

*2.4. Statistical Analyses*

The first series of analyses involved a comparison between classification systems at the lake level; we compared BQIES$_{whole-lake}$ (whole-lake assessment of the BQIES), OECD, LHMS and LHQA with Pearson multiple correlations, using R v 4.0.0 [44], package psych v1.9.2 [45].

The second series of analyses assessed the potential influence of depth on the other environmental metrics, including sediment texture, sediment chemistry, and the water physical and chemical parameters of each single-site sample. Our hypothesis was that the nature of sediment and water features would affect macroinvertebrates, and if the sediment and water features change with depth, space and time, the cascading effects on the macroinvertebrates assemblages could affect the BQIES$_{whole-lake}$ assessment. To verify our hypothesis, we used Generalized Linear Mixed Effect Models (GLMMs [46]), analyzing the effect of depth on the variability of the environmental metrics, accounting

for the pseudoreplication within each lake of the random effects, and for the effect of seasonality as an additional explicit factor in the model. GLMMs were performed with the R package nlme v3.1-147 [47].

After exploring the lake classification and abiotic features of the sampling sites, the main question we wanted to address was the potential confounding effect of sediment and water features in driving the differences in species assemblages of macroinvertebrates in comparison to the representativeness of macroinvertebrates as BQE for the lake quality assessment. We addressed this question by using GLMMs on macroinvertebrates richness, macroinvertebrates SDI and BQIES$_{single-site}$ applied to each site as a function of depth, sediment chemistry, sediment texture, water features and lake ecological status classifications (according to OECD, LHMS, LHQA and BQIES$_{whole-lake}$), accounting for the effect of seasonality and including the effect of the pseudoreplication of sites nested within each lake as a random effect. For the statistical models we used one single explanatory variable summarizing sediment chemistry, sediment texture and water features. To obtain such summary metrics, we performed principal component analyses, one for each group of variables, and kept the first axis as a summary of the metrics of the group.

Before any analysis, dependent variables expressed as percentages were arcsine square root transformed, and dependent variables expressed as count data were log transformed to obtain a Gaussian distribution of residuals [48]. For models including a combination of continuous variables and categorical variables with more than two levels, summary outputs were obtained as type II analysis of deviance tables with the R package car v3.0.7 [49]. Partial r$^2$ for GLMMs were obtained with the R package r2glmm v0.1.2 [50].

## 3. Results

### 3.1. Lake Classification

The range of averaged BQIES$_{whole-lake}$ values from each lake varied between 0.52 (Lake Mergozzo) and 0.22 (Lake Sirio); most of the lakes' estimates were lower than 0.4, a threshold between good (higher than 0.4) and moderate status (Table 2).

**Table 2.** Summary metrics and classification for each analyzed lake, including annual mean hypolimnic oxygen saturation during stratification (O2, %), annual mean total phosphorus at mixing (TP, mg m$^{-3}$), annual mean chlorophyll *a* (Chl *a*, mg m$^{-3}$), and annual mean transparency (m), used to obtain the OECD classification; the other classification scores include BQIES$_{whole-lake}$, LHMS and LHQA.

| Lakes | O$_2$ | TP | Chl *a* | Transparency | OECD | BQIES$_{whole-lake}$ | LHMS | LHQA |
|---|---|---|---|---|---|---|---|---|
| Avigliana piccolo | 3 | 18.56 | 2.17 | 4.17 | oligo-mesotrophic | 0.255 | 26 | 56 |
| Bidighinzu | 3 | 259.16 | 14.67 | 0.89 | hypertrophic | 0.391 | 20 | 47 |
| Candia | 19 | 27.70 | 11.02 | 2.74 | eutrophic | 0.421 | 26 | 56 |
| Liscia | 15 | 41.97 | 6.73 | 3.04 | meso-eutrophic | 0.268 | 18 | 61 |
| Mergozzo | 75 | 4.60 | 1.98 | 7.50 | oligotrophic | 0.518 | 14 | 56 |
| Morasco | 78 | 2.20 | 0.36 | 7.33 | ultra-oligotrophic | 0.464 | 26 | 30 |
| Posada | 32 | 37.44 | 7.35 | 1.97 | eutrophic | 0.370 | 24 | 48 |
| Sirio | 3 | 41.07 | 3.70 | 5.03 | meso-eutrophic | 0.220 | 32 | 53 |
| Sos Canales | 6 | 31.67 | 6.66 | 2.80 | eutrophic | 0.331 | 26 | 52 |
| Viverone | 59 | 80.13 | 3.42 | 5.62 | meso-eutrophic | 0.255 | 32 | 62 |

When classifying Italian lakes on the basis of OECD trophic conditions (Table 2), Lake Mergozzo was the only one with a high annual mean oxygen content even in the deepest layers, low mean concentrations of TP and chlorophyll *a*, and high annual mean transparency values.

The natural lakes of the present study, through the application of LHS (Table 2), were characterized via high habitat quality (mean LHQA ± SD = 56.6 ± 3.3). In two cases (lakes Viverone and Sirio), the lake modification measures (LHMS) were the highest because of the pronounced human littoral alterations. Once again, Lake Mergozzo showed very good hydro-morphological conditions with the lowest LHMS (14) and a quite high LHQA (56) (Table S1), confirming the high habitat quality and

conservation value of its habitats. The reservoirs showed LHMS values never exceeding 20 (Table 2), with the dam representing the only adverse environmental impact.

Each system of lake classification provided a different facet of the assessment of anthropogenic impacts on the habitat. The pairwise correlation values between classification metrics were very low, always below 0.6 as an absolute value, and never significant (Figure 2).

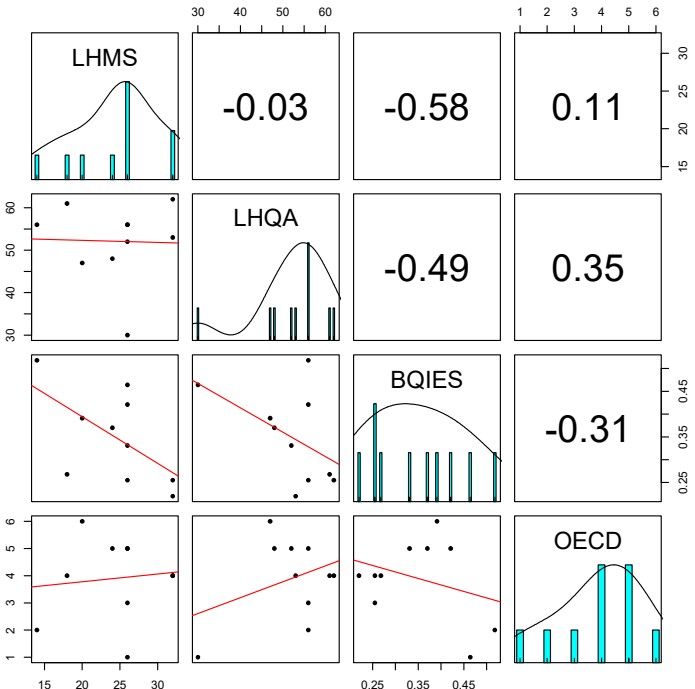

**Figure 2.** Pairwise comparisons between classification systems: LHMS for habitat modification, LHQA for naturalness, BQIES for macroinvertebrates and OECD for trophic status, ranked in order from 1 as ultra-oligotrophic to 6 as hypertrophic. The diagonal reports the histograms of the distribution of each metric; the values above the diagonal represent Pearson's r correlation values; the scatterplots below the diagonal show the correlation between pairs of variables with the (non-significant) trend line.

## 3.2. Single-Site Sediment and Water Descriptors

The sediment chemical components (Table S1) revealed that Morasco reservoir had the lowest average water content (38%) and organic matter percentages (3%); the highest organic matter contents were found in lakes Candia (48%), Sirio (35%) and Viverone (23%), whereas values were <20% in the other lakes (Figure 3). Carbonates were usually present in smaller amounts than the other analyzed components, with values ranging on average from 1% (Sos Canales reservoir) to 20% (Lake Avigliana piccolo).

Sediment texture analyses (Table S1) revealed fine sand as the main component (Figure 3), while silt and clay represented only minor fractions. Morasco reservoir and Lake Candia represented an exception since they showed higher values for both silt and clay. In detail, the lowest fine sand content was found in the Morasco reservoir (60%), and the highest in Lake Sirio (92%). Silt ranged from 6% (Lake Sirio) to 21% (Lake Candia), whereas clay varied from 1% (Liscia reservoir) to 20% (Morasco reservoir).

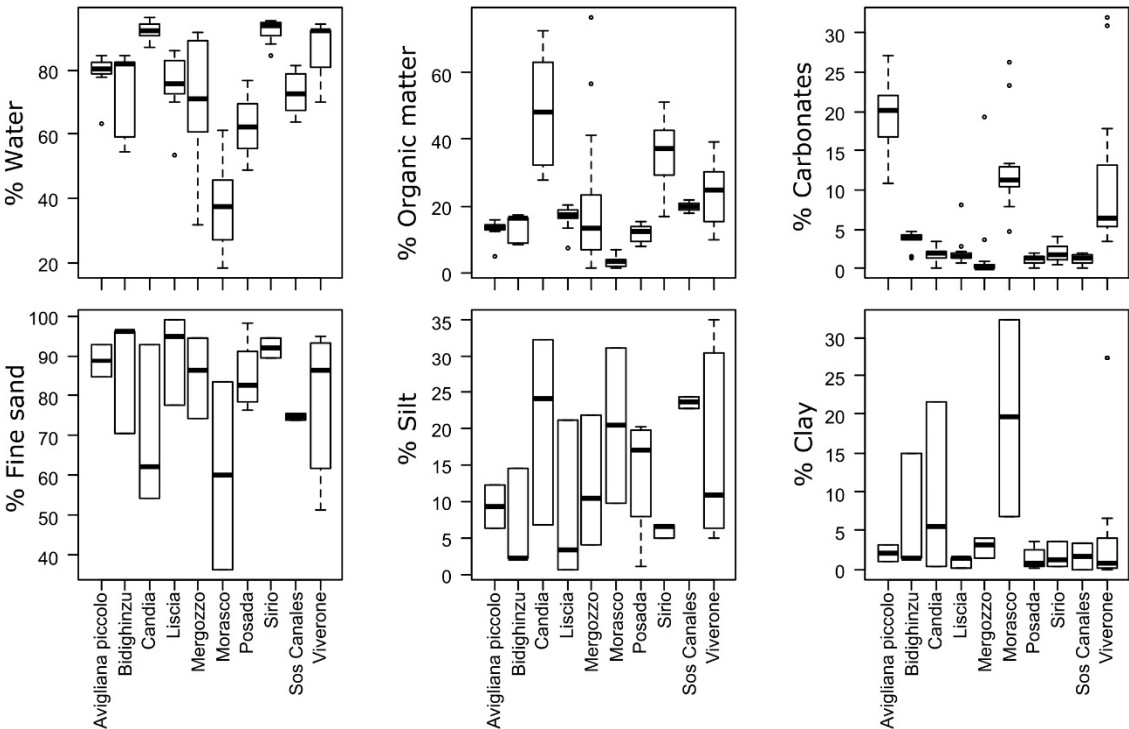

**Figure 3.** Distribution of sediment chemical features (up: water, organic matter and carbonates contents) and soft sediment texture (down: fine sand, silt, clay) for each lake. All variables are reported as percentages.

The water features (Table S1) showed a wide range of conductivity, from 56 (Lake Mergozzo) to >400 $\mu$S cm$^{-1}$ (Lake Avigliana piccolo and Bidighinzu reservoir), and alkalinity (from 14 mg L$^{-1}$ in Lake Mergozzo to 398 mg L$^{-1}$ in Lake Avigliana piccolo). The pH values ranged between 6.5 and 9.1, with the lowest values (<7) in the deep areas of lakes Mergozzo and Sirio and the Sos Canales reservoir, and the highest values (>8) in the littorals of the lakes in north-western Italy. Oxygen saturation was highly variable (from 1.1 to 128%) and strictly dependent on depth and season; the highest concentrations were found along the shores of Lake Sirio, and very low values were found in the deepest layers of Lake Viverone, as well as the Bidighinzu, Sos Canales and Liscia reservoirs, which were close to anoxia (<5%) during prolonged period of water stratification. The nutrient conditions (both TP and TN) showed again a wide range of values: the former varied from 4 $\mu$g L$^{-1}$ (Lake Mergozzo) to 1081 $\mu$g L$^{-1}$ (Lake Bidghinzu), the latter from 2.1 mg L$^{-1}$ (Lake Avigliana piccolo) to >3 mg L$^{-1}$ (Lake Bidighinzu).

Some of the sediment and water features of the samples revealed a significant relationship with depth (Table 3). Among the metrics describing sediment chemistry, only water content was significantly and positively related to depth; among the metrics describing sediment texture, all the percentages of fine sand, silt and clay were significantly and positively related to depth; among water features, temperature, oxygen, pH, TP and TN were significantly affected by depth (Table 3).

**Table 3.** Output of the Generalized Linear Mixed Effect Models (GLMM) assessing the effect of depth on sediment chemistry, sediment texture and water features, including seasonality as a covariate and sites nested within each lake as random effects. Significant effects of depth are marked in bold.

| Environmental metrics | Parameters | Predictor | t-Value | *p*-Value |
|---|---|---|---|---|
| sediment chemistry | water content | (intercept) | 12.6 | <0.001 |
| | | depth | 2.9 | **0.007** |
| | | season | 0.6 | 0.547 |
| | organic | (intercept) | 5.2 | <0.001 |
| | | depth | −1.2 | 0.238 |
| | | season | −0.1 | 0.965 |
| | inorganic | (intercept) | 17.0 | <0.001 |
| | | depth | 1.6 | 0.114 |
| | | season | 1.1 | 0.299 |
| | carbonates | (intercept) | 3.5 | 0.002 |
| | | depth | −1.0 | 0.329 |
| | | season | −2.3 | 0.031 |
| sediment texture | fine sand | (intercept) | 22.0 | <0.001 |
| | | depth | 3.1 | **0.005** |
| | | season | −6.6 | <0.001 |
| | silt | (intercept) | 12.3 | <0.001 |
| | | depth | −2.7 | **0.011** |
| | | season | −5.5 | 0.001 |
| | clay | (intercept) | 3.1 | <0.001 |
| | | depth | −1.6 | **0.019** |
| | | season | −0.1 | 0.598 |
| water features | temperature | (intercept) | 12.5 | <0.001 |
| | | depth | −5.3 | **<0.001** |
| | | season | 3.4 | 0.002 |
| | oxygen | (intercept) | 9.5 | <0.001 |
| | | depth | −3.6 | **0.002** |
| | | season | −1.5 | 0.140 |
| | pH | (intercept) | 49.7 | <0.001 |
| | | depth | −3.1 | **0.005** |
| | | season | −0.1 | 0.940 |
| | conductivity | (intercept) | 5.2 | <0.001 |
| | | depth | 1.5 | 0.142 |
| | | season | 1.5 | 0.146 |
| | alkalinity | (intercept) | 3.3 | 0.002 |
| | | depth | 0.9 | 0.358 |
| | | season | 0.3 | 0.767 |
| | TP | (intercept) | 0.9 | 0.334 |
| | | depth | 2.2 | **0.035** |
| | | season | 0.5 | 0.614 |
| | TN | (intercept) | 4.5 | <0.001 |
| | | depth | 2.4 | **0.023** |
| | | season | −0.9 | 0.361 |

*3.3. Whole-Lake Macroinvertebrates Assemblages*

Macroinvertebrates were represented by 12,799 individuals, belonging to 142 taxa in seven classes (Arachnida, Bivalvia, Clitellata, Gastropoda, Insecta, Platyhelminthes and Malacostraca) and 36 families. Oligochaetes and Chironomids were the most abundant groups, whose overall relative

abundances constituted from 46% (Lake Candia) to 100% (Morasco reservoir) of the macroinvertebrates assemblage (Figure 4).

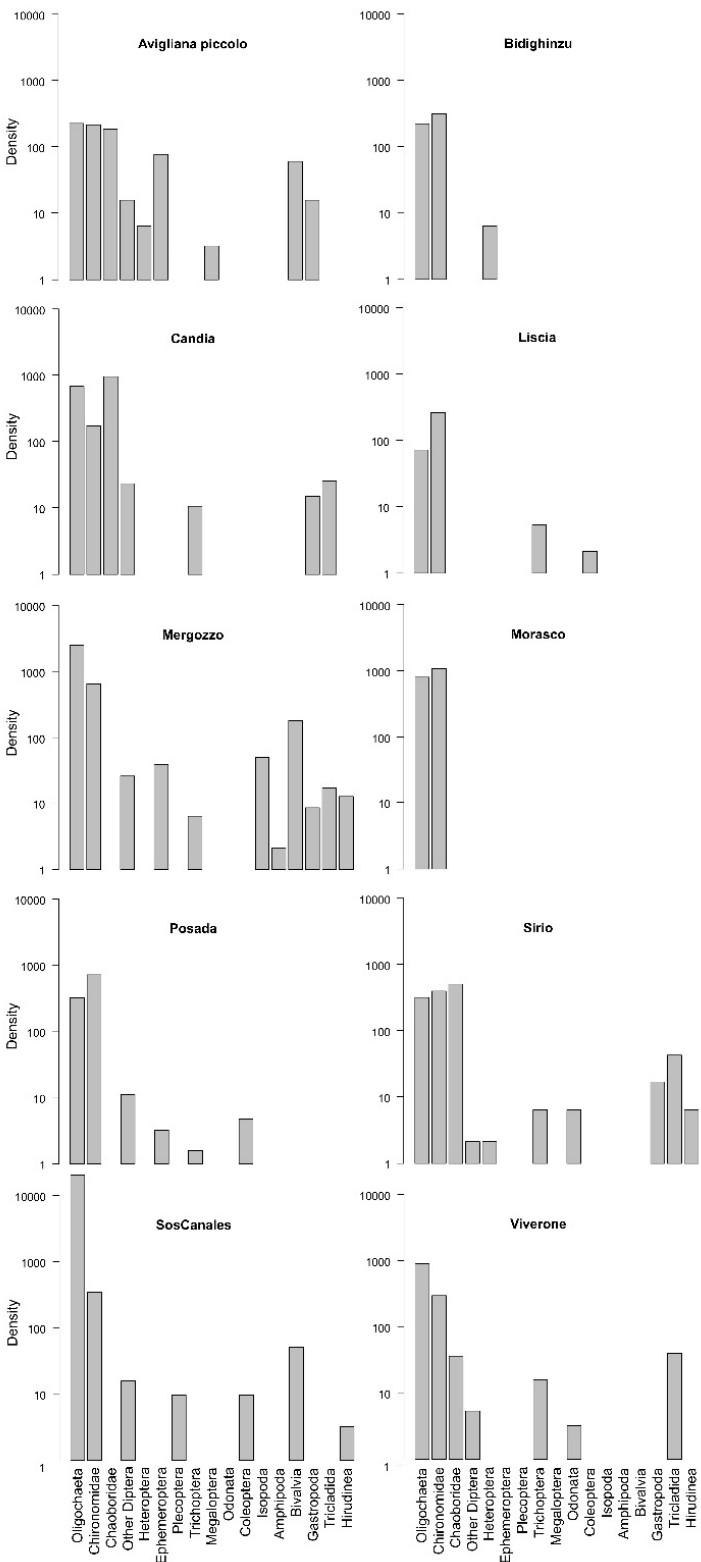

**Figure 4.** Logarithmic (ln) distribution of densities (ind m$^{-2}$) of the main macroinvertebrates taxonomic groups in the different lakes.

Chironomids tended to prevail over Oligochaetes in most lakes, while oligochaetes prevailed in lakes Mergozzo and Viverone, and the Sos Canales reservoir. The latter showed the highest oligochaetes absolute abundances (around 20,000 ind m$^{-2}$), whereas lakes Mergozzo and Viverone had absolute abundances of 2400 ind m$^{-2}$ and 900 ind m$^{-2}$, respectively. Lakes Sirio and Candia presented a large number of chaoborids (550 and 900 ind m$^{-2}$, respectively), while Lake Avigliana piccolo showed comparable densities of oligochaetes, chironomids and chaoborids (225, 216 and 184 ind m$^{-2}$, respectively), and relatively high densities for mayflies, bivalves and gastropods (76, 60 and 16 ind m$^{-2}$, respectively).

### 3.4. Environmental Effects on Macroinvertebrates

Each group of environmental features, namely sediment chemistry, sediment texture and water features, for each site was summarized in one single axis of a principal component analysis, which produced an explained variance 89.8% for sediment chemistry, 77.6% for sediment texture, and 68.3% for water features. These axes were included in the following models as proxies for the three environmental features.

The statistical models to assess the role of potential confounding factors on metrics of diversity (richness of macroinvertebrates, SDI and BQIES$_{single-site}$) revealed that depth had a significant effect, and that, among the metrics of lake quality, LHMS was always significant (Table 4).

**Table 4.** Type II analysis of deviance tables as output of the GLMM assessing the effect of sediment chemistry, sediment texture, water features and depth, including seasonality as a covariate, in addition to the metrics of overall lake quality (OECD, LHMS, LHQA and BQIES$_{whole-lake}$) and sites nested within each lake as random effects, on three metrics of site biodiversity: richness, SDI (diversity) and BQIES$_{single-site}$. Significant effects are marked in bold. The model $R^2$ is reported for each model between parentheses after the name of the response variable.

| Response | Predictor | $\chi^2$ | *p*-Value | $R^2$ |
|---|---|---|---|---|
| | PC axis sediment chemistry | 0.0 | 0.9587 | 0.00 |
| | PC axis sediment texture | 0.7 | 0.3903 | 0.02 |
| | PC axis water features | 0.4 | 0.5486 | 0.01 |
| | depth | 27.5 | **<0.0001** | 0.60 |
| richness ($R^2$ = 0.81) | OECD | 0.1 | 0.7851 | 0.00 |
| | LHMS | 9.5 | **0.0021** | 0.28 |
| | LHQA | 3.8 | 0.0512 | 0.13 |
| | BQIES$_{whole-lake}$ | 0.1 | 0.7200 | 0.01 |
| | season | 5.1 | 0.0768 | 0.04 |
| | PC axis sediment chemistry | 2.0 | 0.1592 | 0.09 |
| | PC axis sediment texture | 0.3 | 0.6050 | 0.01 |
| | PC axis water features | 0.2 | 0.6789 | 0.01 |
| | depth | 24.6 | **<0.0001** | 0.55 |
| SDI ($R^2$ = 0.70) | OECD | 0.6 | 0.4214 | 0.04 |
| | LHMS | 5.3 | **0.0209** | 0.21 |
| | LHQA | 0.0 | 0.8764 | 0.00 |
| | BQIES$_{whole-lake}$ | 0.1 | 0.8197 | 0.00 |
| | season | 3.3 | 0.1964 | 0.04 |
| | PC axis sediment chemistry | 5.5 | **0.0191** | 0.26 |
| | PC axis sediment texture | 0.0 | 0.8685 | 0.00 |
| | PC axis water features | 0.0 | 0.9900 | 0.00 |
| | depth | 20.7 | **<0.0001** | 0.57 |
| BQIES$_{single-site}$ ($R^2$ = 0.78) | OECD | 0.5 | 0.4872 | 0.03 |
| | LHMS | 6.2 | **0.0131** | 0.22 |
| | LHQA | 0.1 | 0.7734 | 0.00 |
| | BQIES$_{whole-lake}$ | 0.0 | 0.8681 | 0.00 |
| | season | 8.3 | **0.0159** | 0.05 |

Depth explained between 55% and 60% of the variance for each model (Table 4), and LHMS between 21% and 28%. None of the other predictors were relevant to explaining richness and SDI, whereas for BQIES$_{single-site}$, sediment chemistry (water content) and seasonality were also significant. Surprisingly, the BQIES$_{single-site}$ scores were not related to the overall BQIES$_{whole-lake}$ scores, nor to the trophic status assessed according to OECD standards (Table 4).

## 4. Discussion

The first unforeseen and positive result of our study at the lake level was that no significant correlations existed between any couple of hydro-morphological, ecological and trophic status assessments (LHQA, LHMS, BQIES$_{whole-lake}$ and OECD) (Figure 2). One explanation could be that each metric assesses a different facet of the environmental impacts of anthropogenic activities, using as they do the descriptive morphologic and hydrologic characteristics of the littorals, the biodiversity of macroinvertebrates or nutrients, and that the different facets of human impacts on different environmental features are not strictly correlated but actually complementary [51,52]. Such differences have two consequences: on the one hand, the different metrics may all be necessary for a reliable implementation of the WFD; on the other hand, ecological assessments cannot be provided with a single number that evaluates lake quality. The Water Framework Directive considers healthy ecosystems to be the basis of sustainable water resources, whereby the various components are interconnected and provide ecosystem services with positive cascading effects on lake resilience to counteract short-term impacts. This means that each assessment, representing different pressures and impacts, did not provide any redundancy in the ecological assessment of lakes, but rather that each of them constitutes a description of the environmental complementarity of the ecological status of each lake. This would have potential implications on mitigation actions to be taken: different metrics of human impacts could already suggest which actions should be considered to minimize impacts on morphological, hydrological, chemical or biodiversity features, understanding their effectiveness in avoiding threats to the provision of ecosystem services on which humanity depends [53].

The other results of our analysis were in line with our expectations: sediment and water features indeed may change with depth, even within the same lake, thus differentially affecting macroinvertebrates assemblages on top of the ecological quality of the lake, and potentially providing biased assessments of lake quality through indices that do not consider such confounding factors. The effect of depth on water and sediment features was not due simply to the fact that the dataset included both deep and shallow water bodies, deeper than 15 m or not; by repeating the same analyses removing the two shallowest lakes (Avigliana piccolo and Candia), the effect of depth was still significant for several water features, namely temperature, oxygen, pH and TN, but not conductivity, alkalinity or TP (Table S2), similar to what happened with the analyses on the overall dataset (Table 3). Regarding sediment, by removing the two shallower lakes, no changes could be seen in the effect of depth on sediment texture and sediment chemistry (Table S2).

The benthic macroinvertebrates species composition in lakes is known to vary with space and time depending on natural factors [54]. As a general rule, environmental variables and biotic interactions influence the macroinvertebrates assemblages, the presence and ratio of sensitive to tolerant species, and the ecological functioning of lakes and their productivity [55–59]. In the present study, the analyses applied to sediment chemistry, sediment texture and water features allowed us to highlight the effect of them, and especially of depth in explaining variations within each lake. This is also in agreement with different metrics of biodiversity, including richness and diversity, confirming the importance of depth in determining the distribution of benthic macroinvertebrates assemblages, and consequently of the BQIES$_{single-site}$, on which basis the BQIES$_{whole-lake}$ is then calculated. Even when removing the two shallowest lakes (Avigliana piccolo and Candia) from the analyses, the effect of depth was still highly significant, and none of the other variables became significant (Table S3). Thus, we can exclude the interpretation that the effect of depth in our study was due to the inclusion of both deep and

shallower water bodies, and the effect holds true even in the analyses including only the subset of deeper water bodies.

The lakes and reservoirs we analyzed presented a fairly similar sediment texture because we focused our monitoring program on soft sediment, and such lack of great variability may explain why no effect of sediment texture was found on the species richness, SDI or BQIES$_{single-site}$ of benthic macroinvertebrates. Watershed characteristics influence ecological activities and equilibria by controlling the chemistry of soils [60], plants [61], waters [62] and microbial community composition [63]. Thus, similar sediments may host communities of benthic macroinvertebrates with similar diversity metrics even if they come from different ecoregions, with different origins of the sediment (e.g., sediment of glacial origin is only found in the Alps). Thus, because of such low variability in sediment texture in the analyzed water bodies, we cannot state whether the inclusion of sediment characterization in terms of the percentage of sand, silt and clay, actually not compulsory within the standardized sampling protocol adopted at national level [26], could be useful when monitoring macroinvertebrates in order to facilitate the interpretation of the results. In our case, the differences in depth overruled any smaller difference in sediment texture, changes in which were also directly correlated to depth (Table 3).

An unexpected result of our analysis was that the BQIES$_{single-site}$, as well as species richness and SDI, were not related to the BQIES$_{whole-lake}$, but were more strongly explained by LHMS, as an index of habitat modification. Thus, the inference we can provide is that macroinvertebrates biodiversity within the lake was affected by visible shore modifications, as evaluated by the LHMS, affecting even the deepest parts of the lakes. Such effect was maintained even when removing the shallowest lakes (Table S3), suggesting that LHMS, contrary to what was found by McGoff and co-authors [64], could indeed provide a reliable metric for the assessment of overall lake quality. To provide some additional speculation in support of the reliability of LHMS as an overall whole-lake assessment, or as a single-site metric, we repeated the analyses using the values of LHMS and LHQA measured for each single-site instead of their whole-lake counterparts, including only the littoral or sublittoral sites in each lake corresponding to the sites where LHMS and LHQA were actually measured. The results revealed that the single-site LHMS and LHQA were never significant, either for richness or for SDI or BQIES$_{single-site}$, and could never explain more than 5% of the variance for each model, with depth remaining the most significant predictor (Table S4). Such additional analyses confirm the validity and reliability of LHMS as a whole-lake assessment of human lake-shore modifications, and their effects on the aquatic macroinvertebrates, regardless of the singe-site measurements for the same index.

## 5. Conclusions

Our results confirm the hypothesis that lake macroinvertebrates assemblages are correlated with confounding factors in each lake, such as sediment chemistry and texture and water features, but that most of the variability can be explained by depth, at least in the set of analyzed water bodies. The main inference that can be suggested from our study is that the BQIES based on the standardized monitoring protocol remains a useful tool for the ecological assessment of the quality of deep Italian lakes, with a mean depth higher than 15 m, as previously suggested [24], but also that, actually, no difference in its results could be highlighted between deeper and shallower lakes. The proposed BQIES cannot be considered definitive, as new species with different auto-ecological requirements could potentially be collected in lakes not sampled yet in the central and southern part of Italy. The indicator weights of species are actually based on a historical dataset of geographic distribution, and need to be updated as monitoring proceeds, slightly influencing the outcome of the BQIES assessment.

Another general inference is the support for the reliability of LHMS as an index of overall lake quality, reflected as a significant predictor in all the analyses of site-related macroinvertebrates biodiversity. This does not imply the abandonment of the BQIES in favor of the LHMS only because the latter was a better predictor of macroinvertebrates community assemblage. Simply, the different metrics represent different pressures with possible divergent scores of quality, and both have to be applied to highlight where to focus management or restoration actions more effectively.

**Supplementary Materials:** The following are available online at http://www.mdpi.com/2073-4441/12/9/2519/s1, Table S1: The Dataset with all the measurements for each single site within a lake, including lake name, site name, season, depth, temperature, oxygen, pH, conductivity, alkalinity, TP, TN, percentage of sand, percentage of silt, percentage of clay, water content, organic matter, carbonates, mineral content, richness of Chironomidae, richness of Oligochaeta, overall species richness, Shannon Diversity Index (SDI), and BQIES$_{single-site}$, LHMS, and LHQA. 'NA' means not available.

**Author Contributions:** Conceptualization, A.B. and D.F.; methodology, D.F.; software, D.F.; validation, A.B., S.Z., R.B., M.C. and D.F.; formal analysis, D.F.; investigation, A.B.; resources, A.B., S.Z. and M.C.; data curation, S.Z.; writing—original draft preparation, A.B., S.Z., R.B., M.C. and D.F.; writing—review and editing, A.B., S.Z., R.B., M.C. and D.F.; visualization, S.Z.; supervision, A.B.; project administration, A.B.; funding acquisition, M.C. All authors have read and agree to the published version of the manuscript.

**Funding:** This research was funded by EU Life + Project INHABIT (Local hydro-morphology, habitat and RBMPs: new measures to improve ecological quality in South European rivers and lakes—Contract No: LIFE08 ENV/IT/000413 INHABIT) under the LIFE + Environment Policy and Governance 2008 program.

**Conflicts of Interest:** The authors declare no conflict of interest.

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
