# Peer review of "The Benthic Quality Index to Assess Water Quality of Lakes May Be Affected by Confounding Environmental Features"

_water, doi:10.3390/w12092519_

Round 1
Reviewer 1 Report
Major points:
- The abstract and introduction mention the comparison between metrics in passing, but to me this is the major contribution of this manuscript. I would bolster the description of the comparisons between the metrics and how they could be used in tandem. For example, it one index suggested low quality and another moderate or good quality, how would those be used to inform management decisions? Are the different indices to be used as multiple lines of evidence for a decision-making process?
- It is a bit unclear from the abstract if depth if a required input in the BQIES. I think this is another main driver of the paper, so emphasize this there. Also in the introduction (lines 92-94).
- If you have enough data, it would be useful to assess the effects of sediment grain size on the BQIES by analyzing a subset of samples that were taken from a similar depth. Because there is not much variation in your sediment type between lakes and reservoirs, it may not matter. If there was more variation across your sites, sediment is likely to also be a significant factor. That is important to consider if lakes in other regions of Italy are not dominated by soft sediment.
- Your conclusion could be stronger based on your findings. Do you think the BQIES should be abandoned in favor or the LHMS if that was a better predictor of macroinvertebrate community assemblage? If you recommend keeping both, what is the benefit of having two metrics and how will you use them, particularly if the scores of quality diverge?
Minor edits:
Throughout – change “sediments” to “sediment” – the latter is used for singular and plural
Throughout – add a comma after “e.g.”
Line 18 – add “of” after “deep layers”
Line 21 – change “textures” to “texture”
Line 38 – replace “strong” with “extensive”
Lines 40-43 – suggested edit: “The WFD is an important component of supporting the water sector in Europe, emphasizing the role of aquatic ecology in management decisions to protect an exhaustible resource.”
Lines 89, 96, 105, 185 – remove “index” (already part of the acronym, BQIES)
Lines 91-92 – change “is affected by” to “incorporates the effects of pollution, as well as physical…”
Lines 92-94 – make it very clear that natural environmental conditions are not required to be incorporated into the BQIES assessments. Suggested change “However, the assessment of natural environmental conditions is not currently required in the application of the BQIES, which may limit its use in comparing across lakes that are assessed by different entities, as well as fail to identify other factors that shape community composition. Here we demonstrate that the nature of sediment, lake depth, and water chemistry—in addition to trophic status—are factors that affect the outcome of the BQIES [28-31].”
Line 96 – change start of sentence to “This paper tests the hypothesis that…”
Line 97 – change “(1 year)” to “(summer to autumn)” or the number of months that spanned your sampling
Line 98 – change “trophy” to “trophic state”
Line 112 – remove “also”
Line 114 – remove “of their” and “of the” from the list – not needed
Line 121-122 – suggested edit: “…allowing us to cover a gradient of trophic states (ultra-oligotrophy to hypertrophy) within which to test our hypotheses.”
Line 123 – replace “are placed” with “are located”
Line 141 – define the method acronyms here instead of in the following paragraph
Line 152 – change “macrophytes” to “macrophyte”
Line 200 – remove comma after “parameters”
Line 209- - remove “to understand”
Line 217 – remove “on summarizing” before sediment texture and water features; not needed
Line 229 – replace “of” with “from”
Line 230 – remove “presented” and edit sentence to “…lake estimates were lower than…”
Table 2 – it would be useful to still have the lake or reservoir label in this table. You reference that difference a lot in the
Line 248 – are all the reservoirs in rural areas? This is not clear from the tables. If they are, you can remove the rural areas wording from the sentence
Line 251 – change to “…habitat; the pairwise correlation…”
Line 267 – change to “Sediment texture analysis (Table S1) revealed…”
Line 277 – change to “Water features showed a wide range…”
Line 305 – remove “of” before lakes
Line 307 – change to “…whereas Lake Mergozzo and Lake Viverone had abundances of 2400 ind m-2 and 900 ind m-2, respectively.
Figure 4 caption – indicate which log base you used
Line 324 – change to “…depth had a significant effect….”
Line 325 – change “depths” to “depth”
Line 328 – move “also” after “were”
Line 329 – change to “Surprisingly, the BQIES scores for single sites were not related to the overall BQIES scope for the whole lake, nor to the trophic status assessed…”
Line 418 – change “may correlate” to “are correlated”
Line 425 – change to “The proposed BQIES index cannot be…”
Author Response
Major points:
- The abstract and introduction mention the comparison between metrics in passing, but to me this is the major contribution of this manuscript. I would bolster the description of the comparisons between the metrics and how they could be used in tandem. For example, it one index suggested low quality and another moderate or good quality, how would those be used to inform management decisions? Are the different indices to be used as multiple lines of evidence for a decision-making process?
REPLY: Thank you for suggesting this important point. We rewrote the abstract by starting explicitly with the comparison in the first sentence and by adding more details of the results related to the comparison. We used the same suggested rationale to change the aims of the study at the end of the introduction.
- It is a bit unclear from the abstract if depth if a required input in the BQIES. I think this is another main driver of the paper, so emphasize this there. Also in the introduction (lines 92-94).
REPLY: actually, it is not compulsory, but as a national representative for the implementation of the activities on lake macroinvertebrates, I always asked to register it to allow us to perform analyses to highlight its importance. After the publication of this paper I can ask (and obtain) a revision of the sampling protocol. In order to remove the doubt, we clarify this issue in the description of the index in the last paragraph of the introduction.
- If you have enough data, it would be useful to assess the effects of sediment grain size on the BQIES by analyzing a subset of samples that were taken from a similar depth. Because there is not much variation in your sediment type between lakes and reservoirs, it may not matter. If there was more variation across your sites, sediment is likely to also be a significant factor. That is important to consider if lakes in other regions of Italy are not dominated by soft sediment.
REPLY: thanks for the suggestion. We already included such analysis in our overall GLMM with .BQIES as a function of depth, sediment chemistry, sediment texture, water features, etc. and sediment texture did not explain any difference in BQIES (Table 4). Thus, we would not discuss the effect of sediment texture. Moreover, given the current use of the BQIES, the index can be applied only to soft sediments similar to the ones we analysed where a grab is used.
- Your conclusion could be stronger based on your findings. Do you think the BQIES should be abandoned in favor or the LHMS if that was a better predictor of macroinvertebrate community assemblage? If you recommend keeping both, what is the benefit of having two metrics and how will you use them, particularly if the scores of quality diverge?
REPLY: We understand the suggestion as a pragmatic solution. Yet, the different metrics we analysed come from rather different empirical data, both abiotic or biotic data. Thus, the output of the different metrics could show different results because they represent different pressures (eutrophication, habitat degradation, and hydromorphological alteration). Thus, our results could be useful to target restoration and management efforts more effectively depending on the single site and on the pressure addressed. We tried to express this idea more clearly at the end of the conclusions section, in order to account for the reviewer’s doubt, which will be potentially common also in other readers.
Minor edits:
Throughout – change “sediments” to “sediment” – the latter is used for singular and plural
REPLY: We would like to thank the reviewer for this comment. We amended the text.
Throughout – add a comma after “e.g.”.
REPLY: corrected as requested
Line 18 – add “of” after “deep layers”
REPLY: we slightly changed the sentence to correct the error
Line 21 – change “textures” to “texture”
REPLY: corrected as requested
Line 38 – replace “strong” with “extensive”
REPLY: corrected as requested
Lines 40-43 – suggested edit: “The WFD is an important component of supporting the water sector in Europe, emphasizing the role of aquatic ecology in management decisions to protect an exhaustible resource.”
REPLY: corrected as requested
Lines 89, 96, 105, 185 – remove “index” (already part of the acronym, BQIES)
REPLY: We thank the reviewer for highlighting this error. We modified also lines 16, 157, 158, 421, and 425 for the same reasons.
Lines 91-92 – change “is affected by” to “incorporates the effects of pollution, as well as physical…”
REPLY: we modified the sentence into “should reflect the effect of pollution on water quality, but also of the physical…”
Lines 92-94 – make it very clear that natural environmental conditions are not required to be incorporated into the BQIES assessments. Suggested change “However, the assessment of natural environmental conditions is not currently required in the application of the BQIES, which may limit its use in comparing across lakes that are assessed by different entities, as well as fail to identify other factors that shape community composition. Here we demonstrate that the nature of sediment, lake depth, and water chemistry—in addition to trophic status—are factors that affect the outcome of the BQIES [28-31].”
REPLY: we modified the sentence adding “but may be used as a further support to the definition of the high ecological status”. In addition, the confusing phrase “natural environmental conditions” has been removed and replaced by “environmental features”.
Line 96 – change start of sentence to “This paper tests the hypothesis that…”
REPLY: corrected as suggested
Line 97 – change “(1 year)” to “(summer to autumn)” or the number of months that spanned your sampling
REPLY: corrected as suggested
Line 98 – change “trophy” to “trophic state”
REPLY: corrected as suggested
Line 112 – remove “also”
REPLY: corrected as suggested
Line 114 – remove “of their” and “of the” from the list – not needed
REPLY: corrected as suggested
Line 121-122 – suggested edit: “…allowing us to cover a gradient of trophic states (ultra-oligotrophy to hypertrophy) within which to test our hypotheses.”
REPLY: corrected as suggested
Line 123 – replace “are placed” with “are located”
REPLY: corrected as suggested
Line 141 – define the method acronyms here instead of in the following paragraph
REPLY: We thank the reviewer for suggesting us how to improve this paragraph.
Line 152 – change “macrophytes” to “macrophyte”
REPLY: we regret we cannot meet the request of the reviewer. The official wording of the quality element here considered is macrophytes
Line 200 – remove comma after “parameters”
REPLY: corrected as suggested
Line 209- - remove “to understand”
REPLY: corrected as suggested
Line 217 – remove “on summarizing” before sediment texture and water features; not needed REPLY: corrected as suggested
Line 229 – replace “of” with “from”
REPLY: corrected as suggested
Line 230 – remove “presented” and edit sentence to “…lake estimates were lower than…”
REPLY: corrected as suggested
Table 2 – it would be useful to still have the lake or reservoir label in this table. You reference that difference a lot in the
REPLY: we regret we cannot meet the request of the reviewer. The sentence of the reviewer is not complete. We do not want to sound repetitive by weighing down a table already rich in data
Line 248 – are all the reservoirs in rural areas? This is not clear from the tables. If they are, you can remove the rural areas wording from the sentence
REPLY: We would like to thank the reviewer for this comment. We modified this sentence as requested and added this info within the paragraph 2.1. Study area at the end of the lakes description.
Line 251 – change to “…habitat; the pairwise correlation…”
REPLY: corrected as suggested
Line 267 – change to “Sediment texture analysis (Table S1) revealed…”
REPLY: corrected as suggested
Line 277 – change to “Water features showed a wide range…”
REPLY: corrected as suggested
Line 305 – remove “of” before lakes
REPLY: corrected as suggested
Line 307 – change to “…whereas Lake Mergozzo and Lake Viverone had abundances of 2400 ind m-2 and 900 ind m-2, respectively.
REPLY: corrected as suggested
Figure 4 caption – indicate which
base you used
REPLY: corrected as requested
Line 324 – change to “…depth had a significant effect….”
REPLY: corrected as suggested
Line 325 – change “depths” to “depth”
REPLY: corrected as suggested
Line 328 – move “also” after “were”
REPLY: corrected as suggested
Line 329 – change to “Surprisingly, the BQIES scores for single sites were not related to the overall BQIES scope for the whole lake, nor to the trophic status assessed…”
REPLY: corrected almost as suggested. We preferred to maintain the BQIESsingle-site and
BQIESwhole-lake wording throughout the text.
Line 418 – change “may correlate” to “are correlated”
REPLY: corrected as suggested
Line 425 – change to “The proposed BQIES index cannot be…”
REPLY: corrected as suggested, but excluding the term index because already part of the acronym as previously suggested.

Reviewer 2 Report
The BQIES index based on the standardized monitoring protocol remains a useful tool for the ecological assessment of the quality of deep Italian lakes, with mean depth higher than 15 m. This study suggest that the depth of each sample should be taken as it currently stands within the prescription of the national standardized protocol. This is helpful in estimating of water quality of lakes.
Author Response
The BQIES index based on the standardized monitoring protocol remains a useful tool for the ecological assessment of the quality of deep Italian lakes, with mean depth higher than 15 m. This study suggest that the depth of each sample should be taken as it currently stands within the prescription of the national standardized protocol. This is helpful in estimating of water quality of lakes.
REPLY: We thank the reviewer for spending time in reviewing our manuscript and for the positive assessment of our work.
